# OpenReview forum: "RootTracker: A Lightweight Framework to Trace Original Models of Fine-tuned LLMs in Black-Box Conditions"
_ICLR.cc/2025/Conference — ICLR 2025 Conference Withdrawn Submission_

### Official Review · Reviewer_tnUe · 2024-11-01

**Soundness:** 2
**Presentation:** 2
**Contribution:** 2
**Rating:** 5
**Confidence:** 3

**Summary:**

This paper proposed RootTracker, a framework designed to identify the original pre-trained model of a fine-tuned LLM through a black-box approach. RootTracker employs a series of prompts and a "knockout tournament" classification mechanism to track the source model of fine-tuned LLMs with an accuracy of 85.7% across several common models.

**Strengths:**

- The studied problem - model tracing - is important for the domain.
- The structure is easy to follow.

**Weaknesses:**

1. Scalability Issues: Current experiments are only conducted with 1.4B parameter models, how would RootTracker adapt or scale to larger models such as 10B or 70B parameters? Would prompt-based classification require modification to retain efficiency?
2. Practical Considerations: Current popular LLMs are all proprietary, where prompt tuning is impossible. Then, will RootTrace be applied to the proprietary setting?
3. Efficiency Issues: while the paper emphasize the efficiency of the method, could you provide any experimental results on the efficiency?
4. Missing Baseline: The paper misses baselines. Could you provide any experiments comparing your method with the baselines?
5. I strongly recommend the authors to further revise the writing of the paper. My suggestions include (1) adding problem setup to clearly describe the problem addressed in the paper and (2) revising the vague sentences in Section 2, e.g., " For instance, if the test model is based on GPT and the classifier is designed to differentiate between GPT and Llama, the classifier’s output would lean toward GPT."

**Questions:**

See Weakness part.

---

> ### Author Response · Authors · 2024-11-22
>
> 1. Regarding scalability, our framework demonstrates that the size of the model does not affect our method's process. Our approach is essentially about identifying corner cases to differentiate between two types of models (similar to fuzzing). Therefore, even with larger models, as long as there are characteristic differences between the models (since different models inevitably have unique features) that manifest differently in certain situations, such corner cases can be identified to distinguish them, thus the performance is unaffected. Additionally, previous work used only 350M models for experiments, and we have significantly increased testing with larger parameter models. Furthermore, testing larger models will greatly increase the demand for computational power, and we plan to enhance our computing capabilities for testing larger models in the future.
>
> 2. Our method involves local adjustments to prompts, so it operates independently of any proprietary settings, not affecting the use of our framework.
>
> 3. Regarding efficiency, firstly, the computational cost of multiple rounds of comparison is not actually that high. With a double-elimination format, for n candidate models, only 2n-2 comparisons are needed, which is a linear increase. Secondly, we store the model's input (prompt) and output (which takes up very little space), and judgments are made based on kNN (which is very fast). Therefore, as the number of models increases, the impact on storage resources and computing speed is minor. Additionally, since training large models from scratch is difficult, there are not many commonly used pretrained (original) models, so a very large candidate pool does not occur. Based on these arguments, our proposed method is highly efficient.
>
> 4. We compared our method with the baseline (in the COMPARISON section of the experiment part).
>
> 5. We will consider adding a "Problem Definition & Formulation" section to describe the issues to be addressed more clearly. Additionally, we will review the terminology used in the paper again.

---

> > ### Comment · Reviewer_tnUe · 2024-11-26
> > **Thanks for the authors' rebuttal**
> >
> > Thanks for the authors' efforts in providing these explanations. However, as most of these rebuttals are based on textual explanation, I would not be more confident about this work before seeing quantitative experiments about (1) the scalability of the framework and (2) the efficiency testing. Moreover, the section 'Problem Definition & Formulation' has not been updated yet. Therefore, I maintain my current score.

---

### Official Review · Reviewer_Bb3M · 2024-11-03

**Soundness:** 2
**Presentation:** 2
**Contribution:** 1
**Rating:** 3
**Confidence:** 4

**Summary:**

This paper presents a scheme for tracing the original models of fine-tuned LLMs. For a given original model, the scheme trains a binary classifier in a sentence embedding space to identify the ground-truth pair of the original and fine-tuned models, among other pairs. The classification and identification are conducted over a set of top-$k$ high-quality prompts and outputs.

**Strengths:**

1. This paper is easy to follow.
2. The experiments include 200 distinct fine-tuned models.

**Weaknesses:**

1. The approach does not address the issues in the motivation. The authors highlighted that fine-tuned models inherit some characteristics from their original models, such as susceptibility to adversarial attacks, potential for jailbreaking, fairness issues, backdoor vulnerabilities, and the risk of generating inappropriate or harmful content. However, telling the original model of a fine-tuned model does not mitigate these harmful characteristics.

2. I do not see an application of the proposed scheme. Watermark approaches can be used if a foundation model team wants to identify fine-tuned LLMs based on their model. If a fine-tuning team or an end user wants to select a good model, directly running evaluation benchmarks can help them identify the optimal model.

3. The prompt selection part is unclear. The authors instruct the original models to generate multiple (50-1000) prompts and select the top $ k$ of them for subsequent sentence classification and model identification. Here, I neither know what the $k$ is (e.g., 10) nor the limitation of publicly available prompts (e.g., Anthropic-HH).

**Questions:**

1. Could the authors provide specific examples of how knowing the original model could help mitigate or analyze vulnerabilities like jailbreaking or fairness issues in fine-tuned models?
2. Could the authors provide a comparison or discussion highlighting the proposed approach's advantages over watermarking and direct evaluation methods in specific scenarios?
3. Could the authors provide the exact value or range of $k$ used in the experiments? Additionally, are there limitations on using publicly available prompts?
4. Suppose the original model is not included in the pairs of models (i.e., not GPT-Neo, GPT-2, TinyLlama, or Pythia as is listed in the experiments), does the scheme produce a false-positive result?

---

> ### Author Response · Authors · 2024-11-22
>
> 1. The reviewer may have misunderstood our motivation. Our method is not aimed directly at addressing issues such as adversarial attacks, jailbreaking, or fairness, but serves as an analytical tool to assist people or other methods in addressing these issues with LLMs. For instance, if we identify that a model is susceptible to adversarial attacks, jailbreaking, or fairness issues, it indicates that other models using the same original model might also have similar problems. Thus, we can utilize our method to identify and locate these models, and then uniformly repair or disable them to prevent greater and more widespread damage.
>
> 2. As described in point 1, the application scenario is intended for third parties and not designed for the foundational model teams. Third parties, such as platforms, forums, or users, might not have access to official watermarks, and officials might not inject watermarks during training.
>
> 3. We propose a framework without specifying fixed values for the adjustable parameters. In different scenarios, these parameters could vary, and we recommend selecting the best-performing parameters based on the situation. In the context of the task, the specific choice of parameters is not crucial to the method's introduction. However, regarding the specific selection of the k-value mentioned in the article, we chose k=5.
>
> 4. Indeed, this framework relies on the information in the model pool. However, it is difficult to expect a model to perform well on unseen data, even if large language models (LLMs) that perform well now do so based on their corpus (known information). Nonetheless, we can improve performance or reduce misinformation through several methods: firstly, by increasing the size of the model candidate pool to include more commonly used models. Secondly, by listing all the models in the model candidate pool and stating that results from models not included in this list are unreliable. Thirdly, by retaining the cosine distance information used in model judgments; if a model's response has a large cosine distance from that of known models, we consider it less reliable.

---

### Official Review · Reviewer_cWBy · 2024-11-04

**Soundness:** 2
**Presentation:** 2
**Contribution:** 2
**Rating:** 3
**Confidence:** 3

**Summary:**

The paper introduces "RootTracker," a framework aimed at tracing the original models of fine-tuned LLMs under black-box conditions. This approach is lightweight and claims efficiency in identifying the base model from a set of prompts using a "knockout tournament" structure. The framework is modular, involving model preparation, database construction, classification, and iterative knockout rounds, enabling the identification of pre-trained model origins without model access. Experimental results report an 85.7% accuracy in identifying the origin models of fine-tuned variants.

**Strengths:**

1.The modular design allows for scalability and future extensions.

2.RootTracker offers a lightweight solution for tracking model origins, potentially requiring fewer computational resources than parameter-tuning methods.

3.The use of a knockout tournament mechanism, inspired by elimination sports, is creative and aims to streamline the comparison process between multiple models.

4.Initial experiments show a high accuracy rate, suggesting the framework’s viability in identifying the base models of fine-tuned variants.

**Weaknesses:**

1.Limited Baseline Comparisons: The framework is compared only to a single prior study, limiting the understanding of its performance relative to other potential attribution methods.

2.Scalability Constraints: The paper reports using models of limited size (up to 1.4B parameters), which might limit RootTracker’s effectiveness when handling models with larger parameter counts, especially in real-world scenarios involving newer, more complex LLMs.

3.Reliance on Predefined Model Pool: The framework's reliance on a predefined set of original models restricts its generalization. If the model under test is not derived from one of these, RootTracker may fail, limiting its practical applications.

4.Computational Demands: Although described as lightweight, the knockout rounds and prompt optimizations still involve iterative comparisons that could become costly with larger model datasets.

**Questions:**

1.How does RootTracker handle cases where the fine-tuned model diverges significantly from any of the original models in the pool?

2.Given the scalability limitations mentioned, does RootTracker plan to accommodate larger models in future iterations?

---

> ### Author Response · Authors · 2024-11-22
>
> 1. As far as we know, we are the second work in this research direction and there are no more baselines, so we used the first work as a baseline for comparison.
>
> 2. Regarding scalability, our framework shows that the size of the model has no impact on the process of our method. The idea behind our method is essentially to find some corner cases that distinguish between two types of models (similar to fuzzing). Thus, even with larger models, as long as the characteristics of the two models are different (different models inevitably have different features), and they perform differently in some cases, it is possible to find such corner cases to distinguish them. Therefore, there is no impact on performance. Additionally, previous work used only 350M models for experiments, and we have significantly increased testing with larger parameter models. Furthermore, testing larger models will greatly increase the demand for computational power, and we plan to enhance our computing capabilities for testing larger models in the future.
>
> 3. Indeed, this framework relies on the information in the model pool. However, it is difficult to expect a model to perform well on unseen data, even if large language models (LLMs) that perform well now do so based on their corpus (known information). Nonetheless, we can improve performance or reduce misinformation through several methods: firstly, by increasing the size of the model candidate pool to include more commonly used models. Secondly, by listing all the models in the model candidate pool and stating that results from models not included in this list are unreliable. Thirdly, by retaining the cosine distance information used in model judgments; if a model's response has a large cosine distance from that of known models, we consider it less reliable.
>
> 4. Firstly, the computational cost of multiple rounds of comparison is not actually that high. With a double-elimination format, for n candidate models, only 2n-2 comparisons are needed, which is a linear increase. Secondly, we store the model's input (prompt) and output (which takes up very little space), and judgments are made based on kNN (which is very fast). Therefore, as the number of models increases, the impact on storage resources and computing speed is minor. Additionally, since training large models from scratch is difficult, there are not many commonly used pretrained (original) models, so a very large candidate pool does not occur.

---

### Official Review · Reviewer_Eq4H · 2024-11-04

**Soundness:** 2
**Presentation:** 1
**Contribution:** 2
**Rating:** 3
**Confidence:** 3

**Summary:**

RootTracker is a framework designed to trace the original models of fine-tuned Large Language Models (LLMs) under black-box conditions. It employs a "knockout tournament" style of comparison to identify the original pre-trained model of a fine-tuned LLM. While the experimental results show a certain level of accuracy, the framework's performance is limited by the predefined pool of candidate models and the high computational cost due to multiple rounds of comparison.

**Strengths:**

Innovative Comparison Method: The "knockout tournament" approach to model comparison is relatively novel in the field of LLM tracing.

**Weaknesses:**

Limitation of Candidate Model Pool: If the base model of a fine-tuned model is not in the candidate pool, RootTracker may not accurately identify it, potentially leading to misleading results.

Computational Overhead: The mechanism of multiple comparison rounds leads to significant computational costs, especially when there is a large number of candidate models.

Diversity of data sets and scenarios: Insufficient discussion of the diversity of data sets and test scenarios used in the paper may affect the universality of the results.

Comparison Efficiency: Multiple comparison rounds may lead to inefficiency, especially when dealing with a large number of candidate models.

Reliability of Results: If the candidate model pool is not comprehensive, the results from RootTracker may be unreliable, limiting its practical application.

**Questions:**

Comprehensiveness of the Candidate Model Pool: How do the authors ensure the comprehensiveness of the candidate model pool, and how do they handle models not in the pool?
Optimization of Computational Efficiency: Have the authors considered optimizing the algorithm to reduce the number of comparison rounds, thereby lowering computational costs?
Improvement of Comparison Method: Is it possible to reduce the necessary number of comparison rounds through other methods, such as machine learning or statistical analysis?

---

> ### Author Response · Authors · 2024-11-22
>
> 1. It is difficult to expect a model to perform well on unseen data, even if LLMs that perform well now do so based on their corpus (known information). However, we can improve performance or reduce misinformation through several methods: firstly, by increasing the size of the model candidate pool to include more commonly used models. Secondly, by listing all the models in the model candidate pool and stating that results from models not included in this list are unreliable. Thirdly, by retaining the cosine distance information used in model judgments (converting cosine distance information to confidence levels). If a model's response has a large cosine distance from that of known models (or a low confidence level), we consider it less reliable.
>
> 2. Firstly, the computational cost of multiple rounds of comparison is not actually that high. With a double-elimination format, for n candidate models, only 2n-2 comparisons are needed, which is a linear increase. Secondly, we store the model's input (prompt) and output (which takes up very little space), and judgments are made based on kNN (which is very fast). Therefore, as the number of models increases, the impact on storage resources and computing speed is minor. Additionally, since training large models from scratch is difficult, there are not many commonly used pretrained (original) models, so a very large candidate pool does not occur.
>
> 3. This article is only aimed at the application scenario of judging original models and does not have high testing requirements for different scenarios. Additionally, as a very new research direction (to our knowledge, we are the second work), there are no datasets available for use (creating datasets would presumably be another task).
>
> 4. Issues related to comparison efficiency have already been explained in point 2.
>
> 5. As described in point 1, and as early work in this direction, our goal is more to provide a way of thinking to solve problems. Regarding the performance of the methods, we will continue to make improvements in future work.

---

### Note · Authors · 2025-07-25

I have read and agree with the venue's withdrawal policy on behalf of myself and my co-authors.

---

### Meta-Review · Area_Chair_yY7g · 2024-12-21

**Metareview:**

"RootTracker: A Lightweight Framework to Trace Original Models of Fine-tuned LLMs in Black-Box Conditions". Based on a database of 1000 prompts generated by API LLMs tasked with generating prompts that would distinguish finetuned models, the submission runs a tournament evaluation of trained pairwise classifiers to evaluate whether a given finetuned model can be traced to a candidate from a small list of base models. Reviewers point out a number of surface-level issues, which the authors partially address, but also note on the limited scalability of the actual evaluation (on only 1B models, and only over candidate sets of size 10), and the limited effectiveness. Even in the restricted setting discussed in the experimental section, the false positive rate of the proposed detector is high.

There are also a number of simple baselines that would have been good to provide, such as established approaches based on analyzing the output of these models in general, such as LLM detectors, or unigram distribution comparisons.

Additionally, the writing of the manuscript could be improved greatly to be made more concise. For example, I entirely do not understand why the submission claims that this scheme is particularly lightweight, modular, and extensible? Similar writing can be found throughout this submission which does not work in the author's favor.

In summary, I do not recommend acceptance of this submission.

**Additional Comments On Reviewer Discussion:**

This metareview was based of my own rereading of the submission, and the reviews of tnUe and Bb3M.

---

### Decision · Program_Chairs · 2025-01-22

Reject